# Insecticidal Efficacy of Microbial-Mediated Synthesized Copper Nano-Pesticide against Insect Pests and Non-Target Organisms

**DOI:** 10.3390/ijerph181910536

**Published:** 2021-10-08

**Authors:** Perumal Vivekanandhan, Kannan Swathy, Adelina Thomas, Eliningaya J. Kweka, Afroja Rahman, Sarayut Pittarate, Patcharin Krutmuang

**Affiliations:** 1Society for Research and Initiatives for Sustainable Technologies and Institutions, Grambharti, Amarapur, Gandhinagar 382650, Gujarat, India; Swathykannan.23@gmail.com; 2Department of Biotechnology, Periyar University, Salem 636011, Tamil Nadu, India; 3School of Pharmacy, Catholic University of Health and Allied Sciences, Mwanza P.O. Box 1464, Tanzania; adelinathomas45@gmail.com; 4Division of Livestock and Human Diseases Vector Control, Tropical Pesticides Research Institute, Arusha P.O. Box 3024, Tanzania; pat.kweka@gmail.com; 5Department of Medical Parasitology and Entomology, Catholic University of Health and Allied Sciences, Mwanza P.O. Box 1464, Tanzania; 6Department of Entomology and Plant Pathology, Faculty of Agriculture, Chiang Mai University, Chiang Mai 50200, Thailand; afroja_r@cmu.ac.th (A.R.); sarayut_pit@cmu.ac.th (S.P.); 7Innovative Agriculture Research Center, Faculty of Agriculture, Chiang Mai University, Chiang Mai 50200, Thailand; 8Research Center of Microbial Diversity and Sustainable Utilization, Faculty of Science, Chiang Mai University, Chiang Mai 50200, Thailand

**Keywords:** *Metarhizium robertsii*, CuNPs, *Artemia salina*, *Eudrilus eugeniae*, *Eudrilus andrei*, *Artemia nauplii*, *Tenebrio molitor*, Mosquitoes

## Abstract

Currently, medical and stored grain pests are major concerns of public health and economies worldwide. The synthetic pesticides cause several side effects to human and non-target organisms. Copper nanoparticles (CuNPs) were synthesized from an aqueous extract of *Metarhizium robertsii* and screened for insecticidal activity against *Anopheles stephensi*, *Aedes aegypti*, *Culex quinquefasciatus*, *Tenebrio molitor* and other non-target organisms such as *Artemia salina*, *Artemia nauplii, Eudrilus eugeniae* and *Eudrilus andrei*. The synthesized copper nano-particles were characterized using, UV-vis spectrophotometer, Fourier Transform Infrared Spectroscopy (FTIR), X-Ray Diffraction (XRD), Energy Dispersive X-Ray analysis (EDaX), High Resolution Scanning Electron Microscope (HR-SEM) and Atomic Force Microscope (AFM) analysis. Insects were exposed to 25 μg/mL concentration produced significant mortality against larvae of *A*. *stephensi*, *A*. *aegypti*, *C*. *quinquefasciatus* and *T*. *molitor*. The lower toxicity was observed on non-target organisms. Results showed that, *M*. *robertsii* mediated synthesized CuNPs is highly toxic to targeted pests while they had lower toxicity were observed on non-target organisms.

## 1. Introduction

Mosquito vectors are distributed abundantly in tropical and sub-tropical regions and transmitting several pathogens to humans and animals. They transmit several vector borne diseases including, dengue, malaria, yellow fever, filariasis and recently zika virus [1]. Malaria is caused by the *Plasmodium* parasite carried by female *Anopheles* mosquitoes, filariasis by female *Culex* and *Anopheles* mosquitoes while dengue, yellow fever and zika are transmitted by *Aedes* pp. [2,3]. *Tenebrio molitor* (Coleoptera: Tenebrionidae) is a stored grain pest [4]. It also damage to several economically important food grains such as, *Zea mays*, *Glycine max*, *Triticum aestivum* [5]. *T*. *Molitor* caused 20% yearly losses in stored grains product in worldwide [6].

Synthetic chemical pesticides such as carbamates, organophosphorus, organochlorines and temephos are mainly to control medical and agricultural pests [3,7]. The repeated usage of chemicals resulted into significant drawbacks and challenges in pest control programme. Thus searching for alternative low cost pesticides, that are eco-friendly, effective and safe for the management of the medical and stored grain pests is essential [8,9,10].

Insect pathogenic fungi are cosmopolitan in nature and also non-virulent to humans and other beneficial organisms [11]. Simultaneously, entomopathogenic fungi have shown to be more effective with minimal dosages to mosquitoes and coleopteran pests [12]. Previous research shown that, *Beauveria* sp., *Metarhizium* sp., *Aspergillus* sp., and *Fusarium* sp., conidia and their chemical constituents have produced remarkable insecticidal activity on different mosquito vectors [2,3,8,9,10,13]. Entomopathogenic fungal derived secondary metabolites conjugated with metal nanoparticles are recommended, are eco-friendly and an improved method for disease vector control [3,14]. Eco-friendly synthesized nanoparticles using botanicals, insect pathogenic fungi and bacteria are cost effective, and target specific for vector control [15]. Copper nanoparticles have a major impact on insect antioxidants and detoxifying enzyme systems. They affect enzymes that are connected to oxidative stress resulting in cell death in insects. In insects, metal nanopesticides bind to S & P protein, DNA and decreases cell membrane permeability resulting into mosquito death [8,16]. *Metarhizium* fungi are one of the most common insect pathogenic fungi; it is soil-borne and infects mainly to medical and agricultural pests that are found in the soil. Recently, *M*. *anisopliae* has been used as microbial pesticides for controlling medical and agricultural pests. Their conidia produced remarkable mortality under laboratory and field conditions [3,17]. There are no toxicological or pathological symptoms observed in birds, fish, mice, rats, and guinea pigs following exposure to *M*. *anisopliae* conidia [18].The entomopathogenic fungi do not cause any risk to humans and the environment [19].

Earthworms are important components of soil ecosystems and are recognized as decomposers in the development of soil macro and micro nutrition. Frequently exposed to synthetic chemical insecticides in the soil, the earthworm will indicate the soil contaminations [20]. Earthworms are also considered as determinants and indicators of soil pollution and fertility [2,3]. They are also an indicator species for various toxicology evaluations [2]. *Eudrilus* sp., are in soils enriched with organic matter [21]. Similarly, Brine shrimp *Artemia* sp.(Anostraca: Artemiidae) are branchiopod crustaceans that tolerate salinity ranging from 4 to 250 gL^−1^. Since they are sensitive to chemicals or other toxicants in aquatic environment, they are widely used for the evaluation of the marine contamination by synthetic chemicals. For example, *A*. *nauplii* and *A*. *salina* are important components of aquatic ecosystems and considered as biomarkers for environmental toxicity [22]. *Metarhizium robertsii* has not been tested for the green synthesis of metal nanoparticles against medical and stored grain pests. In this research, we evaluated the acute toxicity of copper nanoparticles synthesized with *M*. *robertsii* against larvae of *A*. *stephensi*, *A*. *aegypti* and *C*. *quinquefasciatus*, *T*. *molitor* and non-target organisms such as *A*. *salina, A*. *nauplii*. *E. eugeniae* and *E. andrei*. 

## 2. Materials and Methods

### 2.1. Fungal Culture

*Metarhizium robertsii* was isolated from agricultural soils at Sanarappatti, Tamil Nadu, India (12.0933_N, 78.2020_E) (Figure 1A). Insect pathogenic fungi were recovered using the insect bait method with (*Galleria mellonella*) [2,23]. Fifteen early 3rd instar *G**. mellonella* were transferred into a plastic box (5 cm height × 8 cm bottom × 12 cm top) contained 250 g of soil. The soil samples have three replicates which were performed. After transfer of the larvae, the containers were closed with lids and incubated at room temperature (28 ± 2 °C; 85% RH). Each plastic box was examined twice per day for~15 days and dead insect larvae were removed from the containers. They were surface-sterilized for 15 min with 70% ethanol then rinsed in sterile deionized water and then air dried. The insect cadaver was placed on PDA (Potato Dextrose Agar, HiMedia, Maharashtra, India) medium. They were incubated at 28 ± 2 °C; 80% RH for 3 days.

The entomopathogenic fungi was sub-cultured on PDA medium and incubated for 8–12 days at 26 ± 2 °C. The pure culture of insect pathogenic fungi was then incubated in a BOD incubator (Kesar Control Systems, Gujarat, India) at 35 ± 2 °C for further use.

### 2.2. Morphological Confirmation Fungi

The fungal morphological recognition was done based on morphological features such as fungal mycelia color, aerial structures, color making and conidial structure. The fungi were stained with lacto-phenol cotton blue and viewed under light microscope (OlympusCH20i, Mumbai, India) at 40× magnification.

### 2.3. Fungal Broth Culture

The fungal broth was prepared using the method of [8,24]. Cultures were grown on PDA in 250 mL culture flasks. The flasks contained 150 mL of Potato Dextrose Broth (PDB), in which 0.7 g/mL chloramphenicol was added as a bacteriogenic agent. Then 1 × 10^6^ spores/mL was inoculated into the broth. After inoculation, the medium was incubated at 28 ± 2 °C for 12 days.

### 2.4. Genomic DNA Extraction

The fungal immature biomass was cultivated in 50 mL fungi medium (SDB, HiMedia, India). The 1 × 10^9^ conidia/mL was transferred into culturing medium. After, the conidia inoculation media were incubated at 28 ± 1 °C for 6 days. After incubation time, the mycelia were filtered through filter paper and the immature mats were taken for genomic DNA extraction [2,24]. Then, 5 mg of fungi biomass was ruptured in a mortar and pestle with liquid nitrogen. Subsequent to rupturing, 1.5 mL CTAB buffer was added to rupture the mycelia then mixed and transferred to new tubes. The test tubes were moved to a water bath at 60 °C for 40–50 min. After incubation, the 2.5 mL tubes were transferred to 4 °C for 20 min, after which centrifugation at 7000 rpm was performed for 15 min. The aqueous part was transferred to new test tubes, then an equivalent amount of chloroform and isoamyl alcohol (23:2) was made and softly mixed in. The resultant mixture was centrifuged at 13,000 rpm for 12 min, the pellet was removed, and the upper part collected in a separate tube. The same amount (1 mL) of ice-cold isopropanol with 90% ethanol was added to the test tubes and mixed. Then, it was being incubated at 28 °C for 50 min. After incubation, the test tubes were centrifuged at 13,000 rpm for 15 min. Genomic DNA pellets were collected from the centrifuged test tubes and the upper phase was discarded. Genomic DNA was cleaned with 75% ethanol and dried. Purified genomic DNA was taken for further study.

### 2.5. Genomic DNA Purity Confirmation

Insect pathogenic fungal genomic DNA purity was confirmed by using 0.8% agarose gel electrophoresis (HiMedia, Maharashtra, India).

### 2.6. PCR Amplification

The 18s rDNA common primer was used for fungal genomic DNA amplification. PCR reactions were carried out in a 20 µL response volume, which contained 1X PCR buffer with 1.5 mM MgCl, 0.2 mM each of dNTPs (dATP, dGTP, dCTP and dTTP), 1µL DNA, 0.2 µL Phire Hotstart II DNA polymerase enzyme, 0.1 mg/mL BSA, 3% DMSO, 0.5 M betaine and 5 µL of forward and reverse primers. Polymerase chain reaction temperatures were followed in the early stage at 95 °C for 5 min; denaturing at 94 °C for 30 °C and annealing temperature was 50 °C for 30 s; elongation temperature at 72 °C for 2 min and extension temperature at 72 °C for 7 min.

### 2.7. Sequence Analysis

Sequence was checked using a sequence scanner software version1.0 (Applied Biosystem, Sudbury, ON, Canada). Fungal sequence restriction and arrangement were done by using Geneious professional version-5.1.0 software (Applied Biosystem, Sudbury, ON, Canada).

### 2.8. Fungal Aqueous Extract

Fifty grams of fungal biomass was cleaned twice with sterile H_2_O then put in a 250 mL glass beaker that contained 150 mL of sterile distilled water and held for 5 days followed by a cooling extraction [10]. After 5 days, the aqueous secondary metabolites were filtered using Whatman no. 1 filter paper (HiMedia, Maharashtra, India).

### 2.9. Synthesis of Copper Nanoparticles (CuNPs)

Fifteen mL of fungal culture filtrate was transferred to 85 mL of 1 mM copper sulphate solution mixed properly and heated at 60 °C using a hot plate magnetic stirrer (LabFriend India Private Limited, New Delhi, India). After this process, the solution was incubated in dark conditions at 28 ± 2 °C for 72 h and the color changed into a dark brown. Then, the solution was filtered with Whatman no. 1 filter paper, and the solution is washed with double distilled water using centrifugation (Kesar Control Systems, Gujarat, India) with a range at 11,000 rpm for 8 min. The washing measures were repeated many times until the unwanted particles were removed. Finally, the pellet was dried in room temperature for two days and pellets were used for all the experiments and all spectroscopic studies.

### 2.10. Characterization of CuNPs Nanoparticles

Insect pathogenic fungal-derived copper nanoparticles were characterized using UV-Vis spectrophotometer(Cole-Parmer India Pvt. Ltd., Mumbai, India), Fourier Transform Infrared Spectroscopy (FTIR) (PerkinElmer, Waltham, MA, USA), X-ray Diffraction (XRD) (M/S Virtue Meta-Sol, Hyderabad, India), Energy Dispersive X-ray analysis (EDaX) (Jeol, Tokyo, Japan), High Resolution Scanning Electron Microscope (HR-SEM) (Hitachi India Pvt. Ltd., Gujarat, India) and Atomic Force Microscope (AFM) (Nano Science and Technology Company, Uttar Pradesh, India) analysis.

### 2.11. Mosquito Rearing

*A*. *stephensi*, *A*. *aegypti* and *C*. *quinquefasciatus* larvae were obtained from the Institute of Vector Control Zoonoses, Hosur, Tamil Nadu, India and were maintained in the laboratory. Larval bioassays were done at 28 ± 2 °C, 70–80% RH and a 14:10 light–dark photoperiod. Larvae were fed with millet powder [2].

### 2.12. Stored Grain Pests

The *T**enebrio molitor* culture was held in the laboratory at 28 ± 2 °C, 70–80% RH with a 14:10 light–dark photoperiod. The larvae were reared in a plastic container (55 cm long × 38 cm wide × 10 cm height) and fed with wheat bran containing a rich source of macro and micro nutrition. The healthy 4th instar was used for bioassays in the laboratory.

### 2.13. Tenebrio molitor Larval Bioassay

The insect pathogenic fungal-derived CuNPs was determined by calculating the lethal concentration (LC_50_ and LC_90_) values and evaluated with five different concentrations (2–25 µg/mL). Each concentration had three replicates and each replicate had 25, 4th instar *T*. *molitor*. Different concentrations of the fungal-derived CuNPs were applied in a food diet to the larva. The dead *T*. *molitor* larvae were counted 24 h after treatment.

### 2.14. Earthworm Culture

*E*. *eugeniae* and *E*. *andrei* are bio-indicators of soil pollution. They were maintained in laboratory conditions at 28 ± 1 °C [3].

### 2.15. Artemia Species Culture

*A*. *nauplii* and *A*. *salina* larvae were reared in the laboratory in 1000 mL of seawater with a salinity of 30 ppt and the pH was maintained at 8–8.5 with a 17:7 (L:D) photoperiod. The temperature was maintained at 25 ± 3 °C. Aeration was provided every 24 h with an aspirator.

### 2.16. CuNPs Mosquito Larval Toxicity Assay

Larval bioassays were done using the method of the World Health Organization [25]. Five different concentrations from 5–25 μg/mL were tested using 25, 4th instar *A*. *aegypti*, *A*. *stephensi* and *C*. *quinquefasciatus*. For each concentration, there were three replicates and in each replicate, 25 larvae were used. The negative control was used as dechlorinated tap water. After 24 h post treatment, the dead larvae were recorded and probit analysis was done using SPSS 16.00, (IBM, New York, NY, USA).

### 2.17. Artificial Soil Assay

Artificial soil comprised of 15% sphagnum peat, 25% kaolinite clay and 75% fine soil was included for evaluation. This was done according to OECD guidelines with slight modifications [3,26]. A few drops of CaCO_3_ were added to assay to regulate the pH to 6.0 ± 0.5. Soil water content was maintained at 30%. The soil was prepared by adding different concentrations of CuNPs (250 mg/kg) on a dry weight basis. Fifteen adult *E*. *eugeniae* and *E*. *andrei* were transferred to 1000 g of test substrate, each with the desired concentration. The bioassay containers were closed with a lid to prevent the *E*. *eugeniae* and *E*. *andrei* escaping. Thirty days after treatment, the dead *E*. *eugeniae* and *E*. *andrei* were counted. Each concentration had three replicates and each replicate had 15 earthworms. The control was not treated with any CuNPs.

### 2.18. CuNPs’Toxicity on Artemia Species

Mature *A*. *nauplii* and *A*. *salina* were collected with a micropipette and used for toxicity assays with different test concentrations (5–25 μg/mL) of *M*. *robertsii*-derived CuNPs. Sterile seawater was used as a negative control. *A*. *nauplii* and *A*. *salina* LC_50_ and LC_90_ values were calculated at 24 h after treatment and the experiment was replicated 3 times for each concentration. After treatment, the *A*. *nauplii* were observed under binocular microscope for observation of morphological changes.

### 2.19. Histopathological Study

Thirty days after treatment, the CuNPs’ exposed and unexposed *E**. eugeniae* and *E*. *andrei* were separately fixed with 5% formalin for 4 h at 4 °C. Blocks were chilled at 25 °C for 2 h then sliced into 4 μm thinness, with 1.0 mm ribbons, using a microtome (Leica, Germany). The sectioned *E**. eugeniae* and *E*. *andrei* were stained with Ehrlich’s haematoxylin and eosin, and after drying, the slides were observed under a light microscope (OlympusCH20i/India) at 40× magnification.

### 2.20. Ethical Statement

This article does not contain any studies with human participants performed by any of the authors. All applicable international, national, and institutional guidelines for the care and use of animals were followed.

### 2.21. Statistical Analysis

Mortalities of target and non-target species were adjusted using Abbott’s formula, [27]. The experimental dosage was determined by utilizing probit analysis and the values expressed as means ± standard (SE) error of three replicates. The lethal concentrations required to kill 50% and 90% (LC_50_ and LC_90_), and chi-square test were calculated using the SPSS (Statistical Package of Social Sciences) software version 16.0, (IBM, New York, NY, USA).

## 3. Results

### 3.1. Microscopic Confirmation

The microscopic confirmation of insect pathogenic fungi color appearance is light green; the conidial structure was slender in shape. Based on first round morphological structures, the isolated fungi were confirmed as *Metarhizium* species (Figure 1B).

### 3.2. Molecular Confirmation and Phylogenetic Construction

The *M*. *robertsii* genomic DNA purity was identified in 0.8% agarose gel and fungal genomic DNA was amplified using 18s rDNA primer. The amplified DNA size range was 900 bp. The sequence was deposited in the GenBank database (NCBI), and the accession number was MK719963.1. Results of deposited 18s rDNA sequence showed a 100% similarity with *Metarhizium robertsii*. Thus, morphological and molecular studies results confirmed that the isolated entomopathogenic fungi as *M*. *robertsii*. The neighbor-joining tree methods were done for evolutionary confirmation of *M*. *robertsii* (Figure 2).

### 3.3. Synthesis and Confirmation of CuNPs

The *M*. *robertsii* fungal filtrate from CuNPs showed a dark brown color to light blue thus revealing the reduction of 1 mM of copper sulphate (CuSO_4_) to copper nanoparticles (CuNPs) (Figure 3).

### 3.4. UV-Vis Spectrophotometer Analysis

The CuNPs had a strong absorption peak at 670 nm (Appendix A).

### 3.5. Fourier Transform Infrared Spectroscopy (FTIR) Analysis

The FT-IR analysis was done for identification of functional groups in the synthesized copper nanoparticles (CuNPs). The 3441.90 cm^−1^ assigned to ArO–H H bonded, 2831.11 cm^−1^ assigned to –CH2–, 2400.61 cm^−1^ assigned to Ar–CH = CHR. The weaker band at 518.15 cm^−1^ corresponds to S–S disulfide asym in the miscellaneous group Appendix A.

### 3.6. X-Ray Diffraction (XRD) Analysis

XRD shows the crystalline structure and high purity of CuNPs. It had 3 strongest peaks ranging at 29.4389, 35.1523 and 36.8562. The observed strongest peak revealed the crystalline structure of copper nanoparticles (Appendix A).

### 3.7. Energy Dispersive X-Ray Analysis (EDaX) Analysis

Energy-dispersive micro analysis to gain further insight into the CuNPs’ analysis of the sample was performed using the EDaX technique. The peaks ranged from 99.8% indicating the presence of CuNPs (Appendix A).

### 3.8. High Resolution Scanning Electron Microscope (HR-SEM) Analysis

HR-SEM analysis clearly shows an external morphology and crystalline structure of CuNPs and also revealed an exact size ranging from 15.67–62.56 nm (Appendix A).

### 3.9. Atomic Force Microscope (AFM) Analysis

Results of AFM analysis clearly showed the shape of the copper nanoparticles. Most of the nanoparticles were spherical in shape and symmetrical and were also disseminated with no aggregation. The size of the copper nanoparticles was found to be similar in size. AFM imaging technique revealed an X value of 1.5 µm and a Y value of 2.0 nm and a Z value in the golden image of 4.7 nm (Appendix A).

### 3.10. Dynamic Light Scattering and Zeta Potential Studies

Dynamic Light Scattering (DLS) method was used to conclude the nanoparticles’ size distribution. Monochromatic laser diffraction collected by a photomultiplier recorded the polydispersed particles with a size ranging from 100 nm to 500 nm with a Z-average value of 56.13 nm. Zeta potential of synthesized CuNPs was 20.3 mV which confirms the repulsion among the particles and thus increases the stability of the nanoparticles (Appendix A).

### 3.11. Mosquito Larval Bioassay

Green copper nano-pesticide synthesized using *M*. *robertsii* was evaluated for mosquito larvicidal activity against *A*. *stephensi*, *A*. *aegypti* and *C*. *quinquefasciatus*. Results clearly showed that the CuNPs had a strong larvicidal activity against the major mosquito larvae with LC_50_ and LC_90_ values of 3.478–23.717 μg/mL in *A*. *stephensi*, 7.796–65.144 μg/mL in *A*. *aegypti* and 3.136–14.997 μg/mL in *C*. *quinquefasciatus* respectively (Figure 4; Table 1). Among the mosquito larvae, the *C*. *quinquefasciatus* larvae were more susceptible than the other larvae.

### 3.12. T. molitor Larval Bioassay

Copper nano-pesticide synthesized from *M*. *robertsii* was evaluated for larvicidal activity against *T*. *molitor* larvae. The CuNPs had a strong larvicidal activity against *T*. *molitor* larvae with LC_50_ and LC_90_ values of 6.487–29.363μg/mL. (Figure 5; Table 2).

### 3.13. CuNPs’Toxicity on Artemia Species

Green copper nano-pesticide synthesized using *M*. *robertsii* was evaluated for non-toxicity efficacy on aquatic indicators such as *A*. *nauplii* and *A*. *salina*. CuNPs had a lower toxicity to *A*. *nauplii* and *A*. *salina*. The LC_50_ and LC_90_ values were 166.731–450.981 μg/mL in *A*. *nauplii* and 293.901–980.153 μg/mL in *A*. *salina* respectively (Figure 5c; Table 3). The CuNPs induced lower sublethal toxicity to *A*. *nauplii* and *A*. *salina* compared to mosquito larvae and stored grain pests (Figure 5b).

### 3.14. CuNPs’Toxicity on Earthworm and Histopathology

*E*. *eugeniae* and *E*. *andrei* are soil toxicological bio-indicator species. Post treatment, after 30 days, there was low mortality observed in the CuNPs’ treatment. The low sublethal effects of CuNPs were observed in gut tissues. The CuNPs which treated earthworm gut cells were normal, and a regular epithelial surface was observed. There was no cellular debris and the nuclei were round in shape. Similar results were observed in the controls (Figure 6; Table 4).

## 4. Discussion

In this study, the *M*. *robertsii* entomopathogenic fungal culture was isolated using insect bait method [9,23]. The fungi were light green; the conidial structure was slender in shape. Similarly, the conidia were green and also slender in shape as previously reported [2,3]. Based on morphological characteristics, the fungi were confirmed as *Metarhizium* species. Based on the molecular characterization results, the *M*. *robertsii* genomic DNA size range is 900 bp and was amplified using 18s rDNA primer. Similarly, the study in [2] reported that *M*. *anisopliae*-amplified DNA molecule had 1000 bp in size. The *M*. *robertsii* 18s rDNA sequence had100% similarity with *M*. *robertsii*. Based on this similarity, we confirmed the isolated entomopathogenic fungi were *M*. *robertsii*. Vector-borne diseases are considered to be major problems to public health worldwide [2,3,16]. Synthetic chemical insecticide was effective against agricultural and medical pests, but there were challenges faced such as insecticide resistance [17]. Continuous use of chemical insecticides has hampered the insect control programs and formed many insecticide resistance problems [28]. Use of nano-materials in insect control has gained interest because of its eco-friendliness and biocompatibility [10,29].

UV-Vis spectroscopy analysis has the best techniques for preliminary confirmation of the nanoparticles. Copper nanoparticles synthesized from the *M*. *robertsii* culture filtrate had an absorption maxima range at 670 nm, indicating the synthesis of copper nanoparticles. Similarly, Kalaimurugan et al. [30] reported that the pathogenic bacteria *P*. *fluorescens*-YPS3-derived nanoparticles produced an identical range of absorption range. FT-IR analysis was done for identification of the functional groups present in the synthesized copper nanoparticles (CuNPs). The 3441.90 cm^−1^ assigned to ArO–H H bonded, 2831.11 cm^−1^ was assigned to –CH2–, 2400.61 cm^−1^ and assigned to Ar–CH = CHR. The weaker band at 518.15 cm^−1^ corresponds to S–S disulfide asym. This vibrational assignment clearly shows several strong functional groups such as phenols, alkanes, carboxylic acids and aromatics functional groups. Similarly, Logeswaran and others reported a similar kind of functional groups such as carboxylic acids, alkanes, carboxylic acids and aromatics in ethyl acetate crude extract of *G**. applantum*. EDaX peak around 99.8% to the binding energies of copper indicates the reaction product is present in the pure form of copper nanoparticles and also contains a weak signal from the oxygen [31]. The EDaX record for CuNPs’ synthesized fungal culture filtrate showed a strong signal of copper from 2 keV. X-ray emission from carbohydrates/proteins/enzymes presented within the insect pathogenic fungi. A similar level of binding energy was reported in the silver nanoparticles synthesized from the entomopathogenic fungal- and bacterial-derived nanoparticles [8,31].

The results of XRD showed the presence of the crystalline size and shape of the synthesized copper nanoparticles using the *M*. *robertsii* culture filtrate as a reducing agent, and thus confirming the crystallization of the synthesized copper nanoparticles. The entomopathogenic fungi *M*. *anisopliae*-derived silver nanoparticles produced a similar kind of crystalline size and shape of the nanoparticles from the fungal culture filtrates [32]. The particle size was resolute by dynamic light scattering (DLS) dimension. Laser diffraction studies revealed that particle size obtained from highly dispersed mixture was in the range of 15.67–62.56 nm. The zeta potential capacity indicates negative charges (14.35 mV) on the copper nanoparticles. Previous research findings reported similar size range of the nanoparticles from culture filtrate-derived nanoparticles with the size range of 30–150 nm [33].

HR-SEM analysis showed that nanoparticles are spherical in shape and the size ranged from 15.67–62.56 nm. Sarker et al. [33] reported that SEM analysis recorded dissimilar sizes of nanoparticles from the bacteria and fungi culture filtrate-derived selenium particles. Results of AFM analysis characterized the shape. Most of the nanoparticles were spherical in shape, symmetrical and disseminated with no aggregation. The size of the copper nanoparticles was found to be similar in size. AFM imaging technique reveals X value 1.5 µm and Y value was 2.0 nm with Z value in the golden image being 4.7 nm.

Our results on *M*. *robertsii* culture filtrate-derived green pesticides (CuNPs) showed strong larvicidal activity with low LC_50_ and LC_90_ values against *A*. *stephensi*, *A*. *aegypti* and *C*. *quinquefasciatus* mosquitoes, and 4th instar larvae of *T*. *molitor* stored grain pest. Similar reports on entomopathogenic fungi- and bacteria-derived silver nanoparticles produced remarkable levels of mosquito larvicidal activity against 4th instar *A*. *stephensi*, *A*. *aegypti* and *C*. *quinquefasciatus* mosquitoes under laboratory conditions [8,30]. The entomopathogenic fungi *Beauveria bassiana* ethyl acetate-derived chemical constituents produced high damage inside of the *C*. *quinquefasciatus* mosquito larvae [10]; similarly, the combination of insect pathogenic fungal toxins and synthetic chemical constituents produced toxicity against three major mosquito species under laboratory conditions [9].

The 4th instar *T*. *molitor* was highly susceptible to the *M*. *robertsii* culture filtrate- derived green pesticides (CuNPs) at 24 h post treatment under laboratory conditions. Similarly, garlic-derived secondary metabolites/essential oils caused a reduced respiration rate of *T*. *molitor* within 3 h after treatment, probably similar in behavior to their locomotor activity. The respiratory speed and mass of the insect body can stand for the sum of the energy demands of the physiological response of *T*. *molitor* that are essential to create a protection response to chemical constituents [34,35].

CuNPs showed a lower sublethal toxicity to the non-target organisms *E*. *eugeniae*, *E*. *andrei*, *A*. *nauplii* and *A*. *salina*. Green copper nano-pesticides synthesized using *M*. *robertsii* were evaluated for non-toxicity effect on aquatic indicators such as *A*. *nauplii* and *A*. *salina*. Our results showed that CuNPs had a lower toxicity to *A*. *nauplii* and *A*. *salina*. The CuNPs induced lower sublethal toxicity to *A*. *nauplii* and *A*. *salina* compared to mosquito larvae and the stored grain pest. Similarly *M*. *anisopliae* derived ethyl acetate chemical constituent caused lower toxicity and no behavioral changes against *A*. *nauplii* 24 h after treatment [3].

*E*. *eugeniae* and *E*. *andrei* are soil toxicological bio-indicator species. Thirty days after treatment, a low level of mortality was observed in the CuNPs’ treatment. However, there was a low level of sublethal effects of CuNPs that caused tissue damage. The CuNPs- treated earthworm gut cells were normal, regular epithelial surface was observed, there was no cellular debris and the nucleus was round in shape. The same results were observed in the controls. Vivekanandhan and colleagues found similar results from *Metarhizium anisopliae*-derived chemical constituents which produced lower toxic effects on the non-target organism *E*. *eugeniae* 30 days after treatment under laboratory conditions [3].

## 5. Conclusions

Currently, environmental safety is of paramount importance in protection against pollution to guarantee safety for humans and other animals. In the present research, entomopathogenic fungal-derived CuNPs have been found highly useful for controlling mosquito larvae populations in the field. The green synthesized nanoparticles have several advantages such as easily degradable, cheaper and eco-friendly, not toxic to non-target organisms and they are plant-derived. CuNPs can be easily synthesized and used as an effective nano-insecticide for mosquito control. In the present study, the copper nanoparticles were synthesized using *M*. *robertsii*. The copper nanoparticles were tested for their larvicidal activity on 4th instar *A*. *stephensi*, *A*. *aegypti*, *C*. *quinquefasciatus* and *T*. *molitor* and were non-toxic to bio-indicators. Our results show that green synthesized copper nanoparticles (CuNPs) show they are toxic to *A*. *stephensi*, *A*. *aegypti* and *C*. *quinquefasciatus* and less toxic to *E*. *eugeniae*, *E*. *andrei*, *A*. *nauplii* and *A*. *salina*. From this study we concluded that *M*. *robertsii*-derived copper nanoparticles (CuNPs) were safer and not toxic to non-target organisms.

## Figures and Tables

**Figure 1 ijerph-18-10536-f001:**
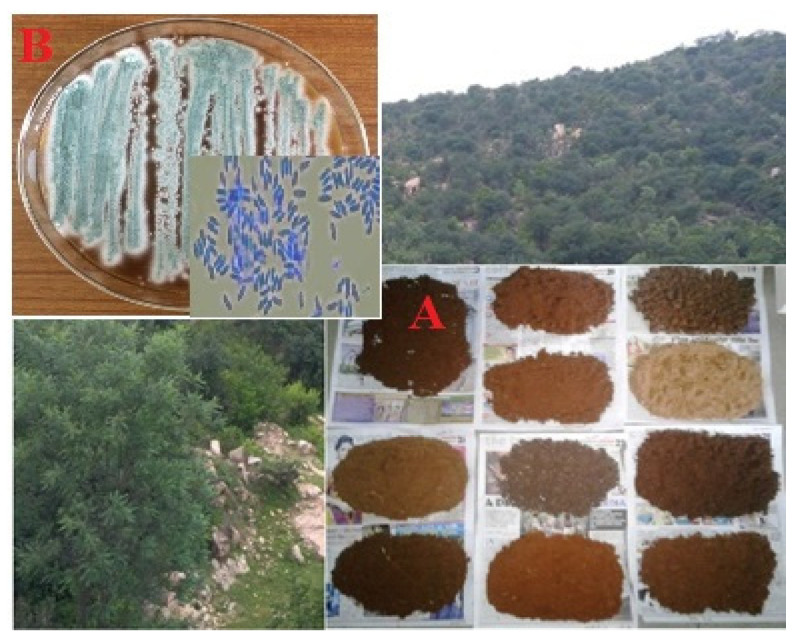
(**A**) Soil collection area and soil sample and (**B**) *Metarhizium robertsii* insect pathogenic fungi.

**Figure 2 ijerph-18-10536-f002:**
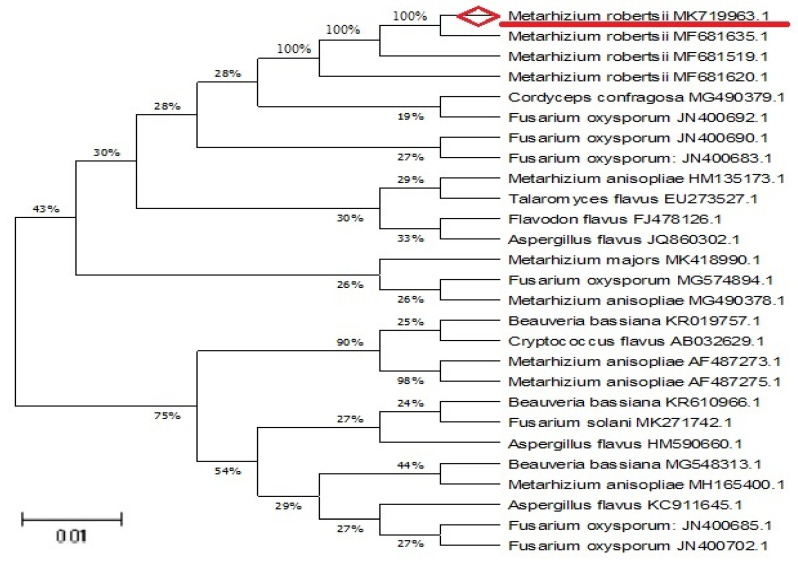
Neighbor-joining tree of the *M*. *robertsii* (MK719963.1) species closely related to the *M*. *robertsii* fungi.

**Figure 3 ijerph-18-10536-f003:**
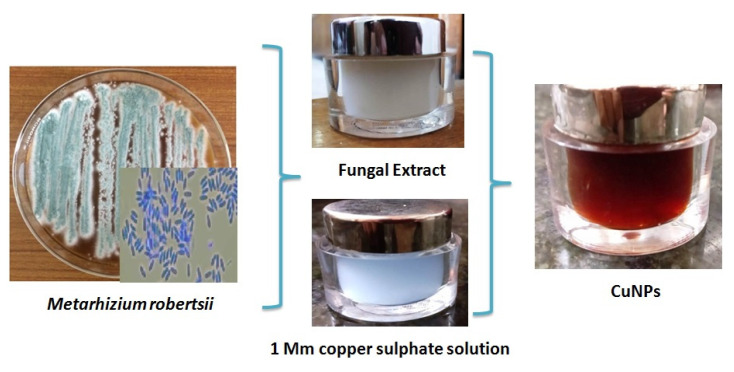
Synthesis of copper nanoparticles.

**Figure 4 ijerph-18-10536-f004:**
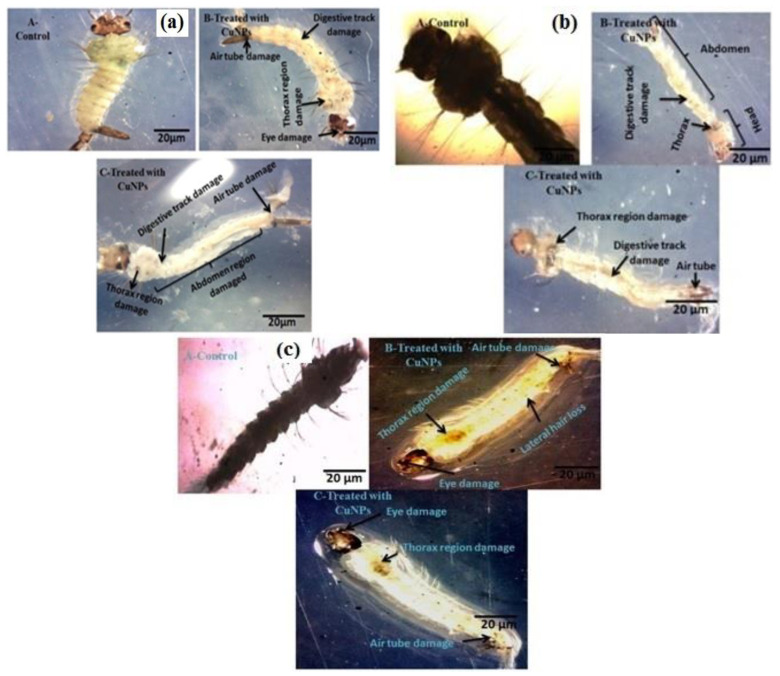
Green nano-pesticide synthesized from *M*. *robertsii* against 4th instar larvae of *C**. quinquefasciatus*, *A**. aegypti* and *A**. stephensi* mosquitoes: (**a**) *C**. quinquefasciatus*, (**b**) *A**. aegypti*, (**c**) *A**. stephensi* at 24 h after treatment.

**Figure 5 ijerph-18-10536-f005:**
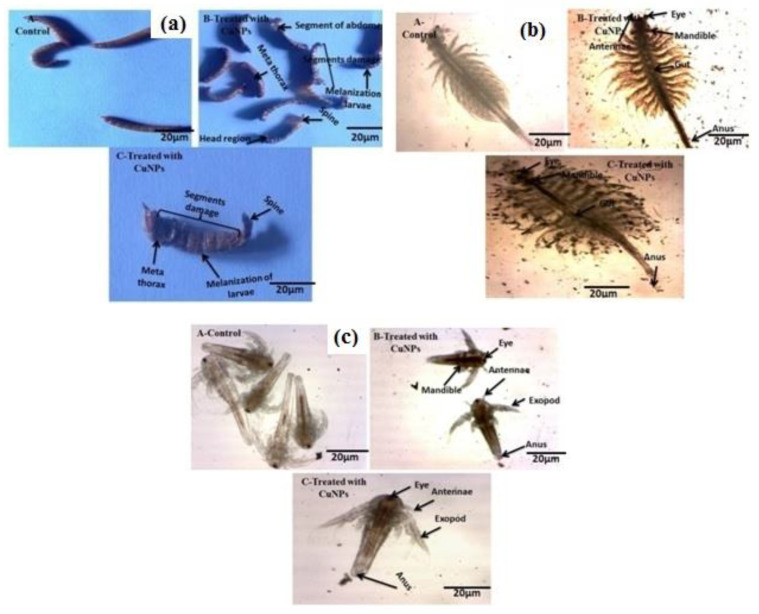
Effect of green copper nano-pesticide synthesized from *M*. *robertsii* against 4th instar larvae of *T*. *molitor*, *A*. *salina* and *A*. *nauplii*: (**a**) *T*. *molitor*, (**b**) *A*. *salina*, (**c**) *A*. *nauplii* at 24 h after treatment.

**Figure 6 ijerph-18-10536-f006:**
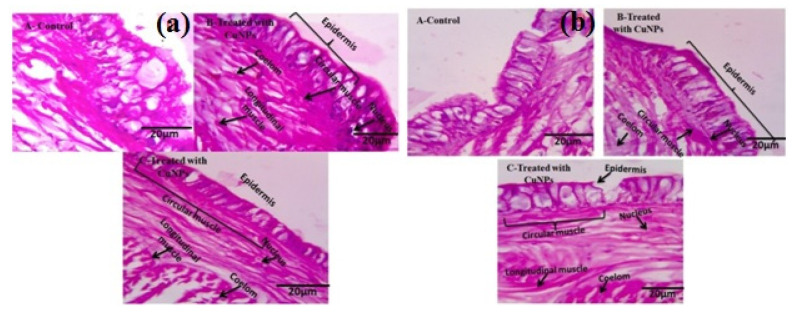
Histopathological studies of green copper nano-pesticide synthesized from *M*. *robertsii* against *E*. *eugeniae* and *E*. *andrei*: (**a**) *E*. *eugeniae*, (**b**) *E*. *andrei* with CuNPs at 24 h after treatment with CuNPs.

**Table 1 ijerph-18-10536-t001:** Mosquitocidal activities of green cluster copper nano-pesticide synthesized from entomopathogenic fungi *M*. *robertsii* against 4th instar larvae of *A*. *aegypti*, *A*. *stephensi* and *C*. *quinquefasciatus* at 24 h after treatment.

Mosquito (na = 450)	Concentration (µg/mL)	% Mortality	LC_50_(LCL-UCL) (μg/mL)	LC_90_(LCL-UCL) (μg/mL)	χ^2^ (df = 12)
*A*. *aegypti*	Control	2.66 ± 0.5	7.796 (5.204–9.886)	65.144 (39.779–188.808)	1.763
	5	42.66 ± 1.0			
	10	52.00 ± 1.0			
	15	62.66 ± 1.7			
	20	70.66 ± 0.5			
	25	80.00 ± 0.5			
*A. stephensi*	Control	1.33 ± 0.5	3.478 (1.569–5.096)	23.717 (17.884–40.603)	3.901
	5	64.00 ± 1.0			
	10	70.66 ± 0.5			
	15	80.00 ± 1.0			
	20	88.00 ± 1.0			
	25	94.00 ± 1.1			
*C*. *quinquefasciatus*	Control	1.33 ± 0.5	3.136 (0.561–6.161)	14.997 (8.967–218.931)	6.915
	5	69.33 ± 1.0			
	10	78.66 ± 0.5			
	15	85.33 ± 0.5			
	20	93.33 ± 0.5			
	25	100.00 ± 0.0			

na = total number of mosquito larvae used per each stage, 25 per replicate, three replicates were carried out, five concentrations were tested; LC_50_ = lethal concentration killing 50% of exposed organisms; LC_90_ = lethal concentration killing 90% of exposed organisms; LCL = 95% lower confidence limits; UCL = 95% upper confidence limits; χ^2^ = chi square; df = degrees of freedom.

**Table 2 ijerph-18-10536-t002:** Effect of green cluster copper nano-pesticide synthesized from entomopathogenic fungi *M*. *robertsii* against *T*. *molitor* at 24 h after treatment.

Insect (na = 450)	Concentration (µg/mL)	%Mortality (µg/mL)	LC_50_(LCL-UCL) (μg/mL)	LC_90_(LCL-UCL) (μg/mL)	χ ^2^ (df = 12)
*T*. *molitor*	Control	0.00 ± 0.0	6.487 (4.746–7.939)	29.363 (22.833–44.577)	1.281
	5	42.66 ± 0.5			
	10	64.00 ± 0.5			
	15	73.33 ± 1.0			
	20	81.33 ± 0.5			
	25	90.66 ± 1.0			

na = total number of *T*. *molitor* larvae used per each stage, 25 per replicate, three replicates were carried out, five concentrations were tested; LC_50_ = lethal concentration killing 50% of exposed organisms; LC_90_ = lethal concentration killing 90% of exposed organisms; LCL = 95% lower confidence limits; UCL = 95% upper confidence limits; χ^2^ = chi square; df = degrees of freedom.

**Table 3 ijerph-18-10536-t003:** Non-target effect of green cluster copper nano-pesticide synthesized from entomopathogenic fungi *M*. *robertsii* against *A*. *nauplii and A*. *salina* at 24 h after treatment.

Organisms (na = 375)	Concentration (µg/mL)	%Mortality (µg/mL)	LC_50_(LCL-UCL) (μg/mL)	LC_90_(LCL-UCL) (μg/mL)	χ2 (df = 12)
*A*. *nauplii*	Control	0.00 ± 0.0	166.731 (117.925–308.963)	450.981 (380.110–490.000)	0.833
	5	34.66 ± 0.5			
	10	40.00 ± 1.0			
	15	45.33 ± 0.5			
	20	53.33 ± 0.5			
	25	58.66 ± 1.1			
*A*. *salina*	Control	1.33 ± 0.5	293.901 (200.295–891.950)	980.153 (890.431–990.001)	1.055
	5	25.33 ± 0.7			
	10	29.33 ± 0.5			
	15	37.33 ± 1.0			
	20	42.66 ± 0.5			
	25	50.66 ± 1.0			

na = total number of adults used per each stage, 25 per replicate, three replicates were carried out, five concentrations were tested; LC_50_ = lethal concentration killing 50% of exposed organisms; LC_90_ = lethal concentration killing 90% of exposed organisms; LCL = 95% lower confidence limits; UCL = 95% upper confidence limits; χ^2^ = chi square; df = degrees of freedom.

**Table 4 ijerph-18-10536-t004:** Non-target effect of green cluster copper nano-pesticide synthesized from entomopathogenic fungi *M*. *robertsii* against *E*. *eugeniae* and *E*. *andrei* at 30 days after treatment.

Earthworm (na = 270)	Concentration (µg/mL)	%Mortality (µg/mL)	LC_50_(LCL-UCL) (μg/mL)	LC_90_(LCL-UCL) (μg/mL)	χ^2^ (df = 12)
*E*. *eugeniae*	Control	0.00 ± 0.0	425.957 (305.941–1296.415)	1091.230 (567.447–1125.151)	0.208
	5	0.00 ± 0.0			
	10	2.22 ± 0.5			
	15	8.88 ± 1.0			
	20	15.55 ± 1.0			
	25	22.22 ± 0.5			
*E*. *andrei*	Control	0.00 ± 0.0	498.178 (330.697–2893.119)	1328.375 (610.925–1395.110)	0.575
	5	0.00 ± 0.0			
	10	2.22 ± 0.5			
	15	6.66 ± 0.5			
	20	8.88 ± 1.7			
	25	20.00 ± 1.0			

na = total number of earthworms used per each stage, 15 per replicate, three replicates were carried out, five concentrations were tested; LC_50_ = lethal concentration killing 50% of exposed organisms; LC_90_ = lethal concentration killing 90% of exposed organisms; LCL = 95% lower confidence limits; UCL = 95% upper confidence limits; χ^2^ = chi square; df = degrees of freedom.

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
