# Peer review of "Insecticidal Efficacy of Microbial-Mediated Synthesized Copper Nano-Pesticide against Insect Pests and Non-Target Organisms"

_ijerph, 2021, doi:10.3390/ijerph181910536_

Round 1

Reviewer 1 Report

Dear Authors,

I have now completed my review of " Insecticidal efficacy of microbial mediated synthesized copper nano-pesticide against insect pests and non-target organisms " and submitted my recommendation, 'Minor Revisions Required'.  I used track changes in the manuscript. Therefore the proposed corrections and remarks can be seen easily.

Final English control is required.

Géza Ripka

Author Response

Reviewer-1

Response to reviewer comments

I have now completed my review of " Insecticidal efficacy of microbial mediated synthesized copper nano-pesticide against insect pests and non-target organisms " and submitted my recommendation, 'Minor Revisions Required'.  I used track changes in the manuscript. Therefore the proposed corrections and remarks can be seen easily.

Final English control is required.

  • Above the correction and language has been corrected by the English expert as per the reviewer comment with track changes mode in the revised manuscript.

Reviewer 2 Report

The main drawback of this paper is poor method part, as now in some cases is not clear amount, proportions of some reagents, temperatures and so on. For exapmle, what kind of heat was used for synthesis of CuNPC? How were samples prepared for all spectroscopic tests?

The company and country of all reagents and equipments should be provided.

Use symbol of degree (not zero) for temperature.

Line 132 - probably mg (not gm).

Latin names in keywords should be in italic.

Clarify/add axis titles in all figures in supplementary file.

Provide relative standard deviation for the data in tables.

Author Response

Reviewer-2

Response to reviewer comments

The main drawback of this paper is poor method part, as now in some cases is not clear amount, proportions of some reagents, temperatures and so on. For exapmle, what kind of heat was used for synthesis of CuNPC? How were samples prepared for all spectroscopic tests?

  • Above the correction has been corrected as per the reviewer comment in the revised manuscript.

=The company and country of all reagents and equipments should be provided.

  • The company and country of all reagents and equipments name has been mentioned in the revised manuscript as per the reviewer comment.

=Use symbol of degree (not zero) for temperature.

  • The symbol of degree has been mentioned for temperature in the revised manuscript as per the reviewer comment.

=Line 132 - probably mg (not gm).

  • The above mentioned typo error gm has been changed into mg in the revised manuscript.

=Latin names in keywords should be in italic.

  • The keywords scientific name has been changed into italic form in the revised manuscript as per the reviewer comment.

=Clarify/add axis titles in all figures in supplementary file.

  • The axis titles in all figures in supplementary file has been clarified as per the reviewer comment.

=Provide relative standard deviation for the data in tables.

  • The standard deviation has been added in the revised manuscript as per the reviewer comments.

This manuscript is a resubmission of an earlier submission. The following is a list of the peer review reports and author responses from that submission.